# Rapid laccolith intrusion driven by explosive volcanic eruption

Jonathan M. Castro[1], Benoit Cordonnier[2,3], C. Ian Schipper[4], Hugh Tuffen[5], Tobias S. Baumann[1] & Yves Feisel[1]

Magmatic intrusions and volcanic eruptions are intimately related phenomena. Shallow magma intrusion builds subsurface reservoirs that are drained by volcanic eruptions. Thus, the long-held view is that intrusions must precede and feed eruptions. Here we show that explosive eruptions can also cause magma intrusion. We provide an account of a rapidly emplaced laccolith during the 2011 rhyolite eruption of Cordón Caulle, Chile. Remote sensing indicates that an intrusion began after eruption onset and caused severe ($>200$ m) uplift over 1 month. Digital terrain models resolve a laccolith-shaped body $\sim 0.8$ km$^3$. Deformation and conduit flow models indicate laccolith depths of only $\sim 20$–200 m and overpressures ($\sim 1$–10 MPa) that likely stemmed from conduit blockage. Our results show that explosive eruptions may rapidly force significant quantities of magma in the crust to build laccoliths. These iconic intrusions can thus be interpreted as eruptive features that pose unique and previously unrecognized volcanic hazards.

[1] Institute of Geosciences, University of Mainz, Becherweg 21, Mainz D-55099, Germany. [2] Department of Geoscience, Physics of Geological Processes, University of Oslo, P.O. 1048 Blindern, Oslo 0316 , Norway. [3] European Synchrotron Radiation Facility, 71 Avenue des Martyrs, CS-40220, Grenoble Cedex 38043, France. [4] School of Geography, Environment and Earth Sciences, Victoria University of Wellington, PO Box 600, Wellington 6140, New Zealand. [5] Lancaster Environment Centre, Lancaster University, Lancaster LA1 4YQ, UK. Correspondence and requests for materials should be addressed to J.M.C. (email: castroj@uni-mainz.de).

Magmatic intrusions are prerequisites to volcanism[1]. Intrusions assemble into magma reservoirs[2], which underpin the fixity and growth of volcanic edifices. If and when magma reaches a volcanic vent it may erupt explosively, effusively or in a combination of both styles[3]. Once an eruption is underway, feedback between magma supply and eruptive flux can impose important top-down control on the migration and evolution of magma bodies, including decompression-triggered magma ascent between subsurface reservoirs[4], or shallow nesting of magma in volcanic edifices resulting in cyclical deformation[5,6].

Understanding volcanic hazards requires deciphering the ties that bind eruptions to their underlying intrusions. Volcano deformation is one of the few surface manifestations of that interaction[7–9], and can indicate how magma supply dictates and responds to the eruption itself[5]. Inflation and deflation cycles of silicic volcanoes[10] reflect the mechanical work done as magma rises into the edifice but becomes obstructed by highly viscous or solidified magma overhead. Under these circumstances, pressures in the shallow conduit may exceed magma chamber overpressure[10], and drive renewed lava extrusion or explosive evacuation of the conduit. Magmatic injections to an erupting volcano are, in essence, short-lived intrusions whose transient overpressure stems from the competition between magma supply and conduit processes[3,11] that produce rheologically strong, low-permeability magma that restricts the outflow of lava.

Here we show that the 'call and response' of magma supply and the eruption can result in growth of laccoliths—shallow, flat-bottomed and concordant magma bodies whose forceful intrusion deforms overlying beds[12]. We use a combination of remote sensing observations, digital terrain models (DTM) and geophysical modelling to constrain key aspects of laccolith emplacement during the 2011 explosive eruption of Cordón Caulle, Chile, the Earth's most recently active rhyolite volcano. Our results indicate that a shallow yet voluminous magmatic intrusion occurred during the early stages of explosive activity, when shallow conduit processes restricted magma output and drove elevated conduit overpressure.

## Results

**The 2011 eruption of Cordón Caulle in Chile.** Cordón Caulle erupted rhyolite magma[13] on 4 June 2011, after about 1 week of elevated seismicity (2–5 km depth; Fig. 1). The VEI 5 eruption ($\sim 1.5$ km$^3$) initiated with pyroclastic columns ($>14$ km) and later produced pyroclastic flows[14–16]. On 7 June, the pyroclastic column became diminished in its height and ejected metre-sized ballistic bombs[14,15]. The bombs are composite breccias with abundant obsidian and pumice fragments, indicating trapping and welding of pyroclastic material under intense shear and gas flux[3]. Scientists identified obstructions in the conduit on 14 June that appeared to restrict the vent, but also prompt continued large bomb blasts[16]. Lava effusion began on 15 June, and occurred at high flux

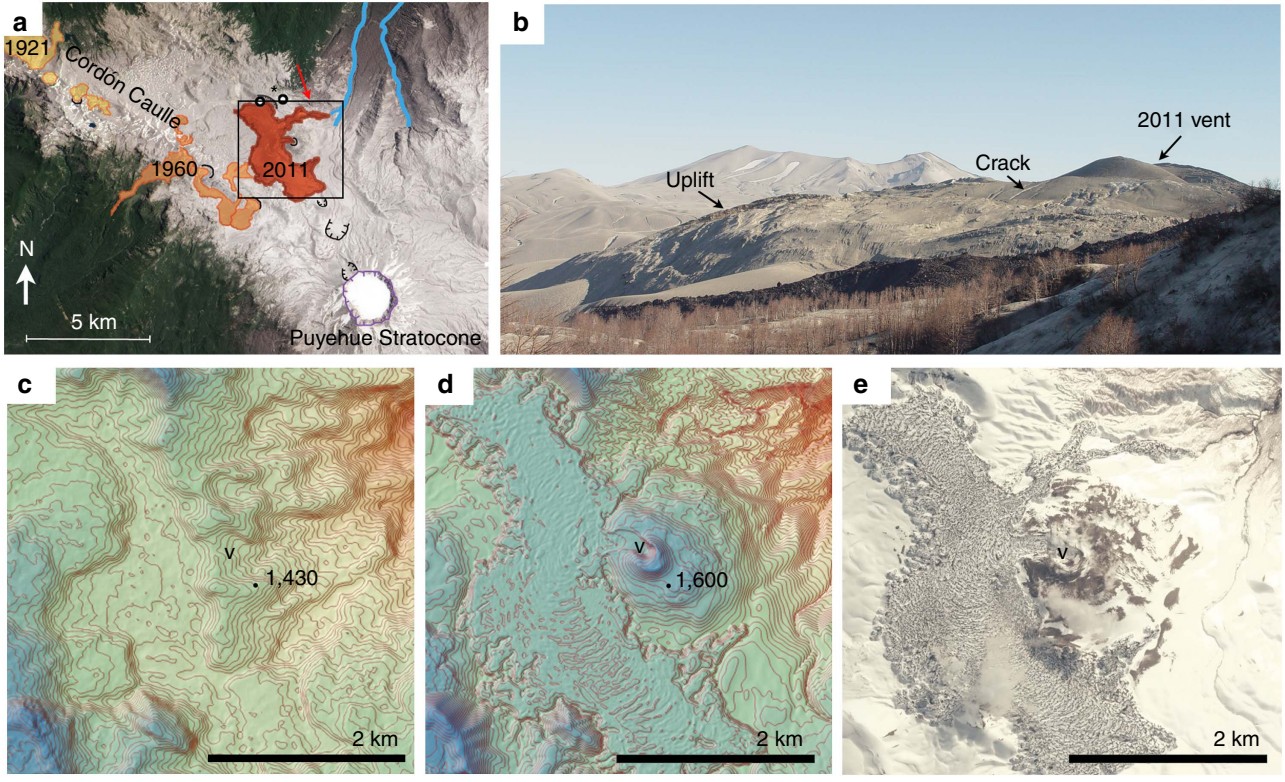

**Figure 1 | Field evidence for syn-eruptive uplift at Cordón Caulle in Chile.** (**a**) NASA image (EO1-ALI; January, 2013) of the Puyehue-Cordón Caulle volcanic complex in Southern Chile (PCCVC) showing recent eruption products. Lava flows are in colour and labelled with their eruption dates. Small black semi circle near centre of the 2011 lava field is the vent. Red arrow indicates the direction of view shown in frame **b**. (**b**) Photograph taken in 2013 of the uplifted surface to the east of the Cordón Caulle vent. The late afternoon shadow accentuates strong uplift amounting to more than 100 m. Puyehue volcano is visible to the southeast. (**c**,**d**) show the pre- and post-eruptive topography (colour-coded and in 10 m contours), respectively, of the area within the black box in frame **a**, as constrained by DTM. Note the location of the vent (v) in the pre- and post-eruptive DTMs. Numbered points indicate the elevations of a particular point and illustrate the amount of syn-eruptive uplift in the vent area. In **e**, a optical satellite image from the winter of 2014 (Bing Imagery) shows that the close correspondence of surface heat flow emanating from the intrusion to the deformation field. Snow-free areas are hotter than the lava flow (Supplementary Fig. 2).

$(20-\sim80\,m^3\,s^{-1})$ (ref. 17), while pyroclastic activity continued for several months[14].

An overflight on 20 June 2011 revealed considerable reduction in the diameter of the active vent, from 400 m in the first week of activity to 50 m (ref. 16). Extensive ground cracking and vigorous degassing around the vent was also observed (Fig. 1b; Supplementary Fig. 1)[11], indicating shallow magma intrusion sometime after the eruption onset (Fig. 1b,c). We constrained when and how this intrusive event happened with satellite imagery, pre- and post-eruptive DTM, and geophysical modelling of the deformed vent area (see the 'Methods' section).

**Remote sensing evidence for laccolith emplacement.** Landsat 7 images indicate no significant change of the landscape on 2 June 2011, just 2 days before the eruption (Supplementary Fig. 1). Thus, deformation must have commenced on 3 June 2011 or after the eruption had started. TerraSAR-X radar images from 6, 8 and 11 June 2011, collected when the eruption was in its purely explosive phase, capture the first signs of deformation, comprising uplifted land to the west of the vent on 8 June 2011, and a linear scarp that grew across the incipient deformation field (Fig. 2a–c). The scarp had disappeared by 6 July, and deformation expanded ($\sim12\,km^2$) and enveloped river valleys northeast of the vent (Fig. 2). The deformation field (Supplementary Fig. 2) did not significantly change in size from about 6 July, implying deformation lasted 1 month ($\sim8$ June to 3 July; Supplementary Fig. 5).

Comparisons of pre- and post-eruption topography (Fig. 1) reveal extreme uplift (tens to hundreds of metres; Fig. 3). Near-vent[18] tephra accumulation ($\sim4$ m; Supplementary Fig. 4)

can account for only a small, localized amount ($<10\%$) of the elevation gain. The $\sim35–60$ m (ref. 19) thick lava flow makes some contribution to total elevation gain, but cannot account for the large DTM-constrained elevation changes within the central part of the flow ($>160$ m; Fig. 3b), because the pre-eruptive basin into which lava flowed was no deeper than $\sim50$ m (Fig. 3c). Parts of the lava-filled basin must have, therefore, been uplifted by as much as 60–80 m (Fig. 3) to account for the observed elevation changes.

The net volume change determined by differencing pre- and post-eruptive DTMs is $\sim1.3\,km^3$ (Fig. 3b). Some of this volume is the lava flow[17] ($\sim0.5\,km^3$), and thus, the remainder comprises the intrusion volume ($\sim0.8\,km^3$). Intrusion over $\sim1$ month (the deformation time) implies an average subsurface magma influx of $\sim300\,m^3\,s^{-1}$.

**Model constraints on syn-eruptive laccolith intrusion.** To decipher the intrusion's radius ($r$), depth ($h$) and pressure ($P$), we apply an axisymmetric deformation model of a growing sill[20] and then compare the modelled deformation to post-intrusive topography. The model[20] applies to laccoliths with a radius-to-depth ratio of $>5$, and that are emplaced as sill-shaped non-compressible fluids that buckle an elastic upper layer after the intrusion reaches a steady diameter[21–23] (Fig. 4a). Lateral injection of magma to build the initial sill may occur along subhorizontal weak planes[24,25] such as those between lavas and pyroclastic deposits beneath the vent (Supplementary Fig. 3).

Figure 4c,d compare modelled deformation and post-intrusion topography. Results are for a possible maximum intrusion pressure of 10 MPa, as estimated from the fragmentation characteristics of Cordón Caulle pumice[11]. The model along A-A' implies $r = 800$ m

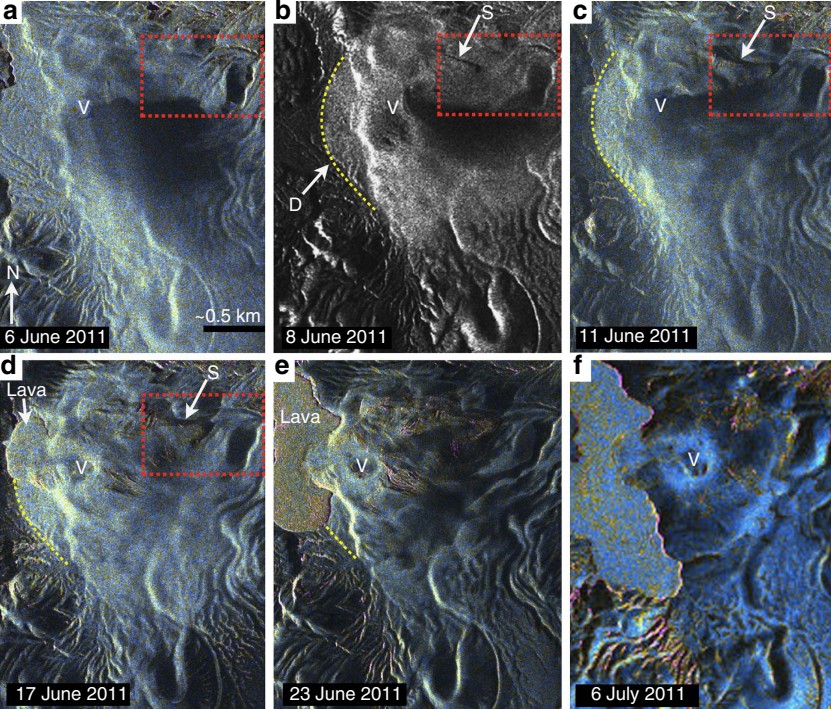

**Figure 2 | Syn-eruptive laccolith growth 6 June to 6 July 2011.** Radar images (TerraSAR-X; supplied by DLR) spanning the onset and duration of laccolith growth. The dates each respective image were collected are indicated in the lower left of each frame. The first three frames (**a–c**) were captured during purely explosive activity, while the last three images (**d–f**) document the early stages of hybrid explosive-effusive activity whereby lava was also emitted with contemporaneous ash plumes[11]. The red-dashed boxes show the areas exhibiting the earliest signs of syn-eruptive deformation, and comprises the formation of a linear scarp structure, 'S', on the 8th of June (see black linear feature) and a broad arcuate area of high radar reflection (indicated by 'D' and outlined in yellow dashes) that can be interpreted as sloping ground. The scarp structure lengthens between the 8th and 17th of June, 2011. Additional evidence for syn-eruptive deformation includes the formation of faults structures (pink lines) adjacent to the vent (V) in frame **e**, and the disappearance of streams located in the eastern extent of the deformation zone (see drainages in centre of red-dashed box in 6 June 2011 image).

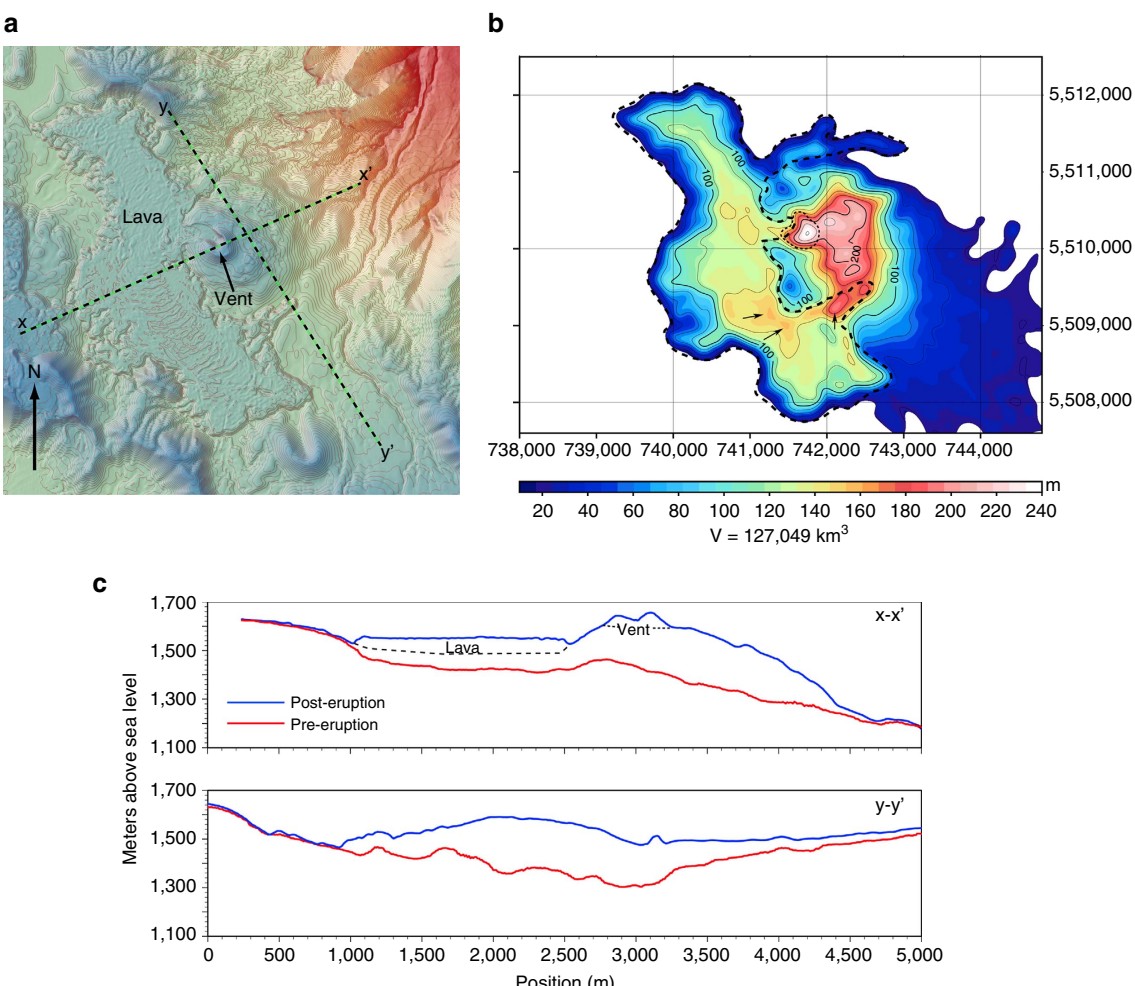

**Figure 3 | Net surficial and volumetric changes from laccolith emplacement.** (**a**) Post-eruptive DTM of Cordón Caulle (March 2014), and (**b**) surface elevation difference map constructed by subtracting the pre- from post-eruption DTMs. The series of colours and contours indicate uplift (m) due to intrusion and lava emplacement. Bold- and thin-dashed lines demarcate the lava flow and tephra cone bounds, respectively. Several areas within the lava flow correspond to anomalously large elevation differences (for example, indicated by orange colour and arrows) that reflect the combination of deformation and lava deposition. (**c**) topographic profile comparisons illustrating the extent and amount of uplift along lines x-x' and y-y'. Contour lines are 10 m apart in **a** and the elevations (m) along profiles are given in **c** on the y-axis. Deformation is concentrated in the eastern portion of the eruption area, however significant uplift (>50 m) is also prevalent beneath the lava flow (arrows in frame **b**), whose expected base is shown by the dashed black line in the upper frame **c**.

and $h = 90$ m, while the fit of the B-B' profile implies $r = 1,400$ m and $h = 195$ m. These solutions are, however, non-unique in that initial intrusion pressures as low as 1 MPa and depths of only tens of metres also match surface topography (Supplementary Fig. 7). On the basis of 200 separate model simulations we establish a range of permissible intrusion depths (20–200 m), and radii (800–2,000 m) for pressures ranging from 1–10 MPa.

The intrusion depths of 20–200 m and the maximum observed uplift amount of $>200$ m indicate an intrusion thickness of $<200$ m, with a base up to 400 m deep. The implied thickness-to-diameter ratio ($\approx 0.13$) along section A-A' is similar to that of exposed natural laccoliths[12,21,26,27] (0.14) and those produced in analogue models[28] (0.1–0.3). The intrusion's form along B-B' is more sill-like. The roughly circular deformation pattern and convex profile (Fig. 3) further suggest a laccolithic intrusion form[26,27].

**Discussion**

Field associations between laccoliths and superposed volcanic deposits are thought to record intrusion followed by eruption[29]. Analogue and numerical models[22,23,27,28] also treat laccoliths

as closed systems that do not exchange mass with their surroundings during their formation. As the Cordón Caulle laccolith formed after the eruption's onset, both magma supply and the eruptive outflow must have influenced its formation. In other words, the Cordón Caulle laccolith was the result of the eruption, rather than its cause. In this sense, the magma supply rate (MSR) must have exceeded the system's magma eruption rate, which in turn, caused overpressure in the conduit sufficient to intrude a sill into surrounding strata.

Overpressure in magmatic conduits will rise in response to a blockage, collapse or diameter reduction as magma accretes along the walls. These phenomena increase flow resistance at the given MSR[30,31]. The onset of Vulcanian blast activity on 8 June and the nearly eight-fold decrease in the vent diameter both manifest conduit constriction at a time when ground deformation was commencing (Fig. 2). Conduit flow simulations (Fig. 5)[31] show that magma pressure beneath the fragmentation level increases as the conduit outlet constricts, reaching a maximum of $\sim 7$ MPa overpressure at 300 m depth when the conduit diameter is reduced by a factor of eight (that is, 400–50 m diameter). These values corroborate the range of laccolith pressures and

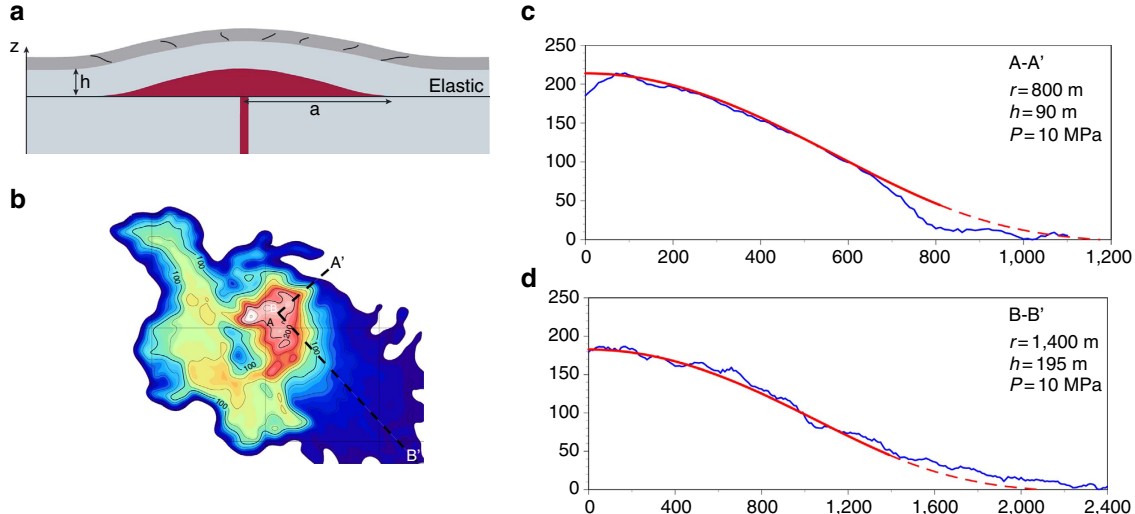

**Figure 4 | Modeling deformation over a growing laccolith. (a)** Model schematic showing a laccolith (red) and the overburden and wall rock in grey colours[23]. The overburden at Cordón Caulle is a mixture of dacite lava flows and pyroclastic fall deposits (Supplementary Fig. 3). **(b)** DTM difference map showing the positions of two surface profiles (A-A′ and B-B′) modelled with the deformation code of ref. 20. **(c)** Comparison of one, non-unique model solution (red) and natural topographic profile (that is, the land surface; blue) along the A-A′ section. The dashed parts of modelled profiles show the uplift occurring outside of the laccolith's diameter. The model fit A-A′ ($R^2 = 0.74$) is compromised by the offset at the position of a growth fault in the natural section. The B-B′ profiles **(d)** are, however, nicely matched ($R^2 = 0.97$).

base depths derived from deformation modelling (1–10 MPa; 220–400 m). We conclude that syn-eruptive conduit constriction at high levels of MSR was a necessary and sufficient mechanism to drive hydrofracturing of neighbouring rocks[32], followed by lateral intrusion of magma to form a sill, and then vertical growth of the Cordón Caulle laccolith (Fig. 5b,c).

The possibility that laccoliths originate from relatively brief, high-intensity eruptions implies that their timescales of formation are far shorter than previously anticipated. The comparably sized Black Mesa laccolith (Utah, USA) is believed to have formed over a period of several decades[26]. Cordón Caulle demonstrates that intrusions of this size can form in a single eruptive pulse lasting just weeks[22]. Such extremely rapid growth is probably unique to syn-eruptive intrusions because explosive eruptions foster high magma flux through the shallow conduit[31]. Whether or not eruption-driven intrusions contribute to the assembly of large crustal reservoirs remains an open question. Short-lived injections to large laccoliths (for example, Torres del Paine, Chile) are below the detection limit of current dating techniques that indicate multi-pulsed growth of over timescales of $\sim 10^4$–$10^5$ years[33]. However, eruption-driven intrusion rates of hundreds of $m^3$ $sec^{-1}$, integrated over recurrence intervals of Cordón Caulle eruptions ($\sim 50$ years) yield similar filling rates ($\sim 0.02$ $km^3$ per year) to those of large laccoliths ($\sim 0.01$ $km^3$ per year) and volcanic reservoirs[1,2,8].

The ability of explosive eruptions to rapidly trigger extremely shallow magma intrusion requires accounting for such intrusion-related hazards at silicic volcanoes. Hazard assessments should consider the long-term effects of hot and voluminous magma shallowly nested beneath the vent, which, as observed at Cordón Caulle in 2013, could fuel unexpected ash and steam explosions (Supplementary Fig. 8). It holds true, however, that eruption-driven intrusions are in and of themselves a form of hazard mitigation, as Cordón Caulle curtailed its own local and immediate hazards by vent blockage and attendant forceful intrusion of nearly 1 km³ of magma. Had this magma erupted, the impacts would have been much greater in both temporal and spatial contexts, with potential for larger and more numerous ash clouds and pyroclastic flows.

## Methods

**Tracking deformation using remotely sensed imagery.** Satellite images of the Cordón Caulle area were used to determine when deformation started and thus whether the laccolith was emplaced pre- or syn-eruptively. We first compared the distributions and geometries of stream drainages around the vent in publically available pre- and post-eruption satellite images to detect changes in the near-vent land surface that might have been related to inflation of the laccolith. To this end, we obtained cost-free images from NASA Landsat 4, 5, 7 and 8 archives and from NASA's Advanced Land Imager (ALI) through the joint U.S.G.S. and NASA download portal 'Earth Explorer'.

We attempted to identify pre- and post-eruptive topographic changes by constructing time-series mosaics of images from the Landsat 4, 5, 7 and 8 campaigns, ideally over time intervals that overlapped the eruption onset (4 June 2011). This was not possible with all of the image series due to cloud cover, the eruption plume blocking the view, or because imaging was not made on particular days of interest. Of the NASA imagery, Landsat 7 provided imaging at a panchromatically adjusted resolution of 15 m per pixel as early as 9 January 2011, and at roughly 2-week intervals until 2 June 2011, just 2 days before the eruption (Supplementary Fig. 1). Landsat 7 images show no deformation through its pre-eruption period of image collection. Landsat 4 and 5 could only document the pre-eruptive land surface on 6 March 2011 and, in these images, we found no signs of deformation in the would-be vent area. Landsat 8 images (acquired in 2013) provided good records of the post-eruptive deformation field and lingering thermal output (sensed by its thermal infrared (TIR) bands), but these images dating from 2013 did not constrain when the apparent uplift had taken place. Two high-resolution orthoimages of the vent area were obtained from Geoeye foundation and from the Astrium company as a means of controlling when the deformation may have ended. These optical images span $\sim 9$ months (3 July 2011 to 10 April 2012) and were the basis of ground deformation maps shown in Supplementary Fig. 5.

Since NASA's optical satellite images were collected infrequently and were sometimes compromised by cloud cover and the eruption column, we resorted to using synthetic aperture radar (SAR) images from the Astrium Company (private partner to the German Space Agency, DLR; Fig. 2) to pinpoint when deformation started. These images provide unobstructed views of the land surface, though in some cases were slightly dithered by the ash in the eruption plume (for example, on 6 June 2011), and depict land surface change as early as 8 June 2011 (Fig. 2). As with the optical satellite images, we identified changes in the position of streams and the appearance of new structures such as cracks or faults to signal the onset of deformation.

Finally, the thermal signature of the deformation zone was tracked using successive Landsat TIR imaging bands, which show the thermal infrared signature of the ground surface (Supplementary Fig. 2). Once the eruption started (4 June 2011), prominent zones of thermal infrared signal were noticed in Landsat 4, 5 and 7 images to the south and east sectors, especially on 26 June 2011 (Supplementary Fig. 2). These hot zones traced out a roughly circular shape in plan view that mimic the form of the deformation field (Supplementary Figs 2 and 5). These hot zones were most prominent in the southwest and northeast where clear scarps with tens of metres of offset grew (Supplementary Fig. 3). These images also show hotspots to the west of the vent where the land was later inundated by the lava flow[17].

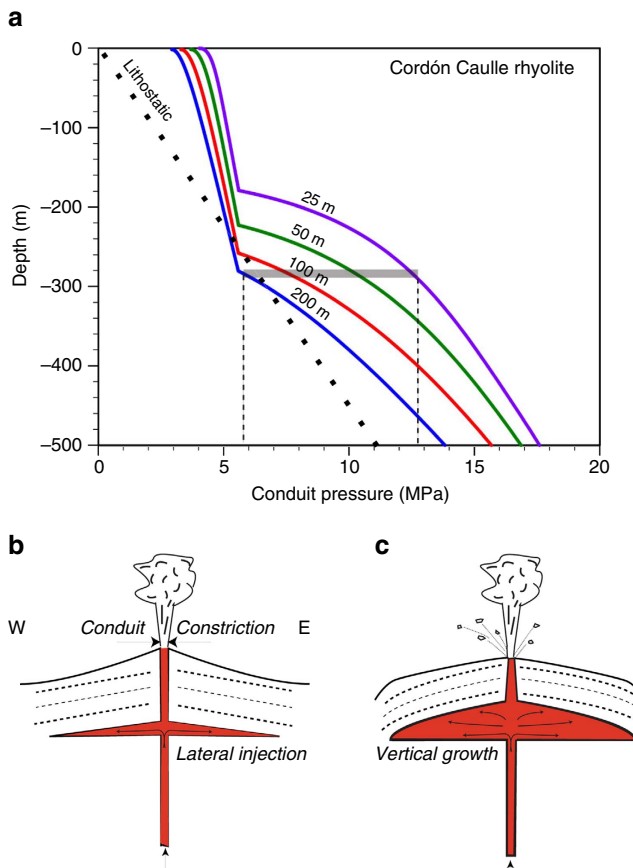

**a**

**b**

W *Conduit* *Constriction* E

*Lateral injection*

*Mass supply*

**c**

*Vertical growth*

*Mass supply*

**Figure 5 | Conduit overpressure as a mechanism to induce shallow intrusion.** Conduit flow simulations[31] showing (**a**) magma pressure versus depth for the Cordón Caulle rhyolite. Lithostatic pressure gradient given as a bold-dashed line. Mass supply rates were in all cases $10^6 \, kg \, s^{-1}$, a value informed by estimates derived from eruption plume of 8 June 2011 and related deposit characteristics[14,15]. Each model curve shows a different magma inlet–outlet radius configuration with the inlet radius held constant (200 m) and outlet decreasing according to the numbers on each graph. Pressure increases as an approximate cube of conduit radius[31] and the characteristic 'kink' in the profiles reflect the fragmentation-induced transition from a bubbly magma to a gas-particle mixture, which in the CONFLOW[31] algorithm occurs at a critical magma porosity of 75%. Constriction of the outlet will cause pressurization of the conduit, in some cases up to ca. 7 MPa (relative to the unrestricted outlet case) at 300 m depth (horizontal grey band). (**b,c**) These images show schematic cartoons featuring the two key steps in syn-eruptive laccolith formation. Sill formation (**b**) must occur before the laccolith can inflate and deform overlying strata (**c**).

**Digital terrain model.** We used high-resolution digital elevation data purchased from the Astrium company to quantify the uplift of the post-eruption land surface in a 100 km² area around the vent. A post-eruption DTM was constructed using stereo optical images dated 13 March 2014 and collected by the Pléiades satellite constellation. The DTM has a lateral and vertical resolution of 4 m. We assessed the amount of uplift at Cordón Caulle by subtracting the ASTER global DTM, which predates the eruption (April 2011), from the Pléiades DTM, using Generic Mapping Tools and QGIS software. This operation created a difference map from which net topographic and volume changes could be assessed. Some of the land surface and volume differences between the two DTMs represents deposition of tephra and lava, but our field observations (Supplementary Fig. 4) indicate relatively modest tephra accumulations near the vent (0.5–4 m thickness), which corroborates results of other studies of the airfall deposit[14,15,18] and indicates that the effective volume change estimates inferred from the DTM subtraction are not significantly influenced by the fall of tephra (that is, tephra is <10% of the total uplift amount). The contribution to

the volume change by lava was assessed by field measurements of lava flow front thicknesses and the aerial distribution of lava in satellite optical and radar images[17]. We checked for systematic vertical offsets between the pre- and post-eruption DTMs by comparing far-field topography profiles; vertical offsets between the two DTMs are typically <10 m (Supplementary Fig. 6).

**Deformation modelling.** We used the numerical code of Galland and Sheibert[20] to simulate ground deformation above an inflating axisymmetric sill and infer, from the model solution fits to natural topography, the intrusion pressure, size and depth. The model, implemented with Matlab R2014b for Macintosh, generates 2D deformation profiles that are readily comparable to the natural surface topography. Thus, we used two orthogonal directions in the deformed region (Fig. 4) as modelling targets to estimate the laccolith's diameter, pressure and depth. We investigated the effect of profile position on the resulting solutions and found that the shortest and longest dimensions of the deformation field produced the biggest differences in laccolith depth (~overburden thickness) and radius; oblique sections produced intermediate values of depth and size compared with the two extreme sections (A-A' and B-B' of Fig. 4).

Modelling the deformation above a growing sill or laccolith relies on many physical parameters whose values are not known *a priori* or whose possible variations result in a large range in model solutions for the targeted unknowns of intrusion size, pressure and depth. The elastic roof thickness ($h$) in the model is the thickness of the rigid layer above a growing laccolith and does not include material that behaves in a granular fashion (Fig. 4a). Therefore, the depth estimates derived from deformation models are minimum values because some unknown amount of the overburden is unconsolidated granular tephra. The primary model inputs are the Young's modulus of the elastic roof ($E$), elastic roof thickness ($h$), sill radius, sill pressure distribution and elastic foundation stiffness ($k$). Of these parameters, the Young's modulus ($E$) of the upper elastic layer is highly important because it imposes a strong first-order control on the resultant deformation (uplift) for given values of the other parameters. For example, for fixed intrusion diameter and depth, changing the Young's modulus by an order of magnitude results in about 5 MPa of intrusion pressure variation. We used Young's moduli of $10^9$–$10^{11}$ Pa to account for the abundant tephra and andesite lava units, respectively[34] in the basement[35] (Supplementary Fig. 3). The deformation model also requires an input for the elastic foundation stiffness ($k$), a poorly constrained parameter ranging from about $10^4$ to $10^9$ Pa m$^{-1}$ (ref. 22) that limits the amount of deformation outside of the intrusion radius, and results in a more bell-shaped profile with increasing foundation stiffness. We examined a range of possible $k$ values and found it possible to mimic the shape of the natural topography profiles for $k$ ranging from $10^5$ to $10^7$ Pa m$^{-1}$. The pressure distribution in the intrusion is not known *a priori,* but we assume it to be uniform[22] (model parameter $n = 10$) and depth estimates are based on several parameters whose possible ranges of values are large, and therefore this introduces some uncertainty in the model results. To minimize this uncertainty, we used ranges of values of Young's modulus[34], elastic foundation thickness and a constant pressure profile. Taken together, no particular combination of variables produced the best match (Supplementary Fig. 7) to the natural topography, but rather a number of non-unique best-fit solutions exist. Supplementary Fig. 7 shows the results of some 200 independent model simulations whereby the Young's modulus and elastic foundation thickness were varied. The overburden density (2,000 kg m$^{-3}$) was kept constant in all simulations.

Clearly, the Galland and Scheibert[20] model simplifies many natural complexities that may underpin errors in intrusion parameters. We know, for example, that the intrusion is oblong, and formed under irregular topography (Figs 3 and 4), whereas the model requires an axisymmetric geometry and uniformly thick overburden. It is unclear whether the intrusion's asymmetry would imply lower- or higher-than-normal magma input pressures. The variability in overburden topography is a likely source of error, because the sloping cover to the eastern half of the intrusion (Fig. 3b) would have prompted more uplift and consequently higher inferred pressures. The model, furthermore, does not incorporate faults, as observed along the intrusion's northeastern edge (Supplementary Fig. 3). Faults alleviate stress resisting buckling and allow more uplift for a given magma pressure. An offsetting effect to faulting is the presence of exsolved gas in the magma[13], as this highly compressible phase would dampen some of the energy of intrusion. Owing to these complexities, we consider the pressure and depth estimates from the two profiles to define a range of permissible values.

**Conduit flow modelling.** To assess the overpressure that could be generated by conduit constriction, we apply the CONFLOW model[31] to simulate the Cordón Caulle rhyolite magma flowing (900 °C) (ref. 13) through a 1-km-long tapered conduit at a MSR equal to the magma eruption rate on 8 June (~$10^6$ kg s$^{-1}$; Fig. 5) (ref. 15). The model accounts for the effect of conduit constriction by allowing a reduced outlet diameter on an otherwise cylindrical conduit.

**Data availability.** All relevant data in this manuscript are available from the authors.

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

## Acknowledgements

We are grateful for the constructive comments from reviewers Helge Gonnermann, Steffi Burchardt and Alessandro Bonforte. We thank Mats Landgren and the Geoeye foundation for providing imagery of Cordón Caulle. J.M.C. was supported by the VAMOS research centre at the University of Mainz. C.I.S. acknowledges Victoria University FSRG grant number 205424, a Royal Society of New Zealand Cook Fellowship awarded to C.J.N. Wilson. H.T. is supported by a Royal Society University Research Fellowship.

## Author contributions

The manuscript was co-written by all authors while the data was sourced, processed, and interpreted by Jonathan Castro.

## Additional information

**Competing financial interests:** The authors declare no competing financial interests.

**Publisher's note**: 

