## [Peer Review File · Nature Communications]

Reviewer #1 (Remarks to the Author):

A. Summary of the key results

In their manuscript, Castro and coauthors report on surface doming during the 2011 Cordon Caulle eruption, which they explain by the rapid emplacement of a subvolcanic laccolith. The presented study gives evidence for a case where an intrusion starts to form immediately after the eruption onset, and continues while the eruption is ongoing.

B. Originality and interest

The syneruptive emplacement of a laccolith is an interesting observation, but what really caught my interest was the interpretation that laccolith intrusion was driven by the eruption. The latter is in a way opposite to what is commonly assumed, namely that intrusion precedes eruption. Hence, the manuscript describes an intriguing case of syneruptive intrusive activity, a process that could not be understood from the fossil geological record of a laccolith. As such, this study is appealing to a broad readership in the Earth sciences and makes a valuable link between volcanology and intrusion studies.

C. Data and methodology

Remote sensing data on surface doming is corrected for the amount of eruptive product accumulation and modelled using an analytical model by Galland and Scheibert (2013). The input parameters and uncertainties in both input and model results are presented in a clear and understandable way. The best-fit solution is a laccolith intrusion at 20 to 200 m depth. The magmatic overpressure assumed in the models is corroborated by calculations of the pressure increase due to conduit constriction during the eruption. The overall approach appears valid and is presented in a transparent way.

Some of the figures are a bit small. I suggest to increase their size and the font size of labels. I also think that the Supplementary Fig. S5A should be added to the main manuscript text with some small adjustments (see below).

D. Appropriate use of statistics and treatment of uncertainties

The range of fitting models is described and uncertainties are discussed.

E. Conclusions

The conclusions are valid and the transparent presentation of the uncertainties allows the reader to evaluate the robustness of the results and conclusions.

As the manuscript ends with the discussion (apart from the Methods section), I suggest to add a summarising sentence, a take-home message of some sort.

F. Suggested improvements

I have listed some minor comments in the table below.

Line Comment

43 Please replace precursor by prerequisite, as a precursor is a signal to most volcanologists.

45 The intrusion will often create a new vent, please mention

52 I guess what the authors mean is not "rigid magma", but the host rock or similar. Please correct.

64 Please replace "m's" by e.g. "metre-sized"

73 The fractures mentioned in the text are hard to identify in Fig. 1. They are, however, nicely represented in the supplementary figure S5. I suggest adding Fig. S5A to Figure 1 and increase the thickness of the lines or the overall size of S5A. I find this satellite image a lot clearer to understand. The area of uplift in relation to the location of the vent is easier to see.

107 Please add some references to articles on sill emplacement that support the suggestion that the initial sill was emplaced along a weak plane, e.g. Kavanagh et al., 2006, EPSL; Burchardt, 2008, JVGR.

117 Change "m's" to "m".

124 If the format of Nature Communication articles allows, I suggest to add a Discussion headline

here to separate results from Discussion.

165 Which part of the Supplementary Information does this statement refer to? It is unclear whether the authors indicate that the 2013 eruption was fed by the laccolith. More explanations are needed here or in the Supplementary Information.

I also suggest to add a reference to Sigmundsson et al. 2010 Nature on how a shallow sill was hit by a dyke resulting in the explosive part of the Eyjafjallajökull eruption in 2010.

Figures Please consider increasing the size of most of the figures and the fonts within the figures.

G. References

I suggest to add some references. See my comments under F.

H. Clarity and context

The abstract is clear and appropriate. Introduction and conclusions are well written. I suggest to add a final statement (see above).

Regards,
Steffi Burchardt

Reviewer #2 (Remarks to the Author):

The manuscript presents the discovery of shallow magma intrusion (laccolith) during the 2011 eruption of Cordon Caulle volcano in Chile. The work is based on remote sensing observations of surface deformation and of thermal emission, together with field observations. Overall this is an interesting discovery/hypothesis that is suitable for publication in Nature Communications. I therefore recommend the manuscript for publication. While I see no major flaws with the work presented, I do have some comments for consideration.

1.) Figure 4: In my opinion it should be shown that a laccolith provides indeed the better match to the observed deformation compared to other deformation sources. For example, one could consider inflation of the upper conduit as a null hypothesis that needs to be ruled out. This should be easy to do, as there are analytical models for pipe-like inflation sources. This could be discussed within one or two sentences in the manuscript, with a brief discussion of the analysis deferred to the supplementary information.

2.) Figure 1E: Could an alternate interpretation of this image be that snow has accumulated in local topographic lows (troughs) and that higher (more exposed) areas, such as small ridges/swells or steep slopes, have less/no snow cover? If that is not a viable alternative, perhaps add a brief sentence explaining why that is so?

3.) Figure 2: These images are not easy to interpret. It would be helpful if the images were larger in size and if there was more annotation. What is the date on the left panel of Figure 2? It would be helpful to add a time line to this figure, indicating what the eruptive activity was. This would allow the reader to place the images into the overall context of the eruptive progression.

4.) I like Figure 3. Figure 3A is identical to Figure 1D and perhaps redundant, given likely space constraints. When I studied Figure 1, I was hoping for a difference map between 1D and 1C, which is shown in Figure 3C. I am wondering if it would make sense to combine Figures 1 and 3?

5.) Figure 5: I understand the objective for doing this modeling, but wonder whether it is actually necessary for the purpose of this manuscript. In fact, it potentially raises some detracting questions. For example, in Figure 4 it seems that the "best" model is for an intrusion depth of 90 m. According to Figure 5A that would seem to be above the level of fragmentation and imply intrusion of tephra. There are also a number of other questions about the conduit modeling that

need to be given consideration, as there are a number of assumptions and uncertainties associated with such modeling (conduit size/shape, inlet pressure, exit conditions, magma rheology, volatile content, etc.). These, together with an assessment of uncertainty/sensitivity of model predictions, as well as a table of model parameters and parameter uncertainties, need to be presented in the supplementary information. Alternatively, one could perhaps defer a more thorough discussion of such modeling to a separate paper. In my opinion the paper does not require such modeling to make a convincing case for syn-eruptive intrusion.

6.) Figure S7: It seems to me that the deformation modeling is the central part of this paper, because it is THE hypothesis test. Although not a necessary requirement, I am wondering whether it would not be sensible to provide a more in-depth presentation of the results of this analysis within the manuscript itself?

Specifically, what I find lacking is an adequate presentation of model tradeoffs. For example, one could provide 3D graphs or 2D contour plots showing goodness of fit as a function of radius and pressure, as well as depth and pressure, and radius and depth for a given Young's modulus. This would indicate how well the model is resolved.

7.) How is model "fit" defined?

8.) Figure 4C: How is "natural topographic variations" defined? Is that the post-intrusion topographic profile minus the pre-intrusion profile?

Helge Gonnermann

Reviewer #3 (Remarks to the Author):

This is an interesting paper dealing about a particular and outstanding deformation measured during the eruption at Cordon caulle volcano, revealing a shallow subsurface dynamics of magma accompanying the eruptive event. Authors present a multidisciplinary investigation using data coming from different techniques. Data are well presented and discussed and the interpretation of the phenomenon fits the observation at surface. Authors perform also a model calculation even if, as they clearly and correctly state, the conditions are at the limits of the model assumptions; mainly due to the very shallow intrusion and the clearly not-elastic condition of the deformed medium. Personally, I think that a model is not always necessary for demonstrating an observed phenomenon, especially when the observations are so outstanding and the conditions are not so close to the theoretical assumptions; sometimes, the uncertainties of the model are bigger than those associated to a simple direct interpretation by experts. In this case, however, "signal-to-noise" ratio is very high and this allows reliable results and Authors correctly show the dispersion of the parameters of 200 model processings. More interesting is (in my opinion) the conduit simulation in order to understand the pressure and depth of the intrusion process.

Manuscript is well written, I had only some difficulties in following lines from 78 to 85, trying to making order on the dates and the evolution of the deformation. Maybe a summary table or a simple timeline showing the most important dates of the observed phenomena and the images acquisitions could be useful to the reader.

After this, i think that this paper deserves to be published.

Alessandro Bonforte

I. Reviewer 1 (Steffi Burchardt) comments:

A. Summary of the key results

In their manuscript, Castro and coauthors report on surface doming during the 2011 Cordon Caulle eruption, which they explain by the rapid emplacement of a subvolcanic laccolith. The presented study gives evidence for a case where an intrusion starts to form immediately after the eruption onset, and continues while the eruption is ongoing.

No action required.

B. Originality and interest

The syneruptive emplacement of a laccolith is an interesting observation, but what really caught my interest was the interpretation that laccolith intrusion was driven by the eruption. The latter is in a way opposite to what is commonly assumed, namely that intrusion precedes eruption. Hence, the manuscript describes an intriguing case of syneruptive intrusive activity, a process that could not be understood from the fossil geological record of a laccolith. As such, this study is appealing to a broad readership in the Earth sciences and makes a valuable link between volcanology and intrusion studies.

No action required.

C. Data and methodology

Remote sensing data on surface doming is corrected for the amount of eruptive product accumulation and modelled using an analytical model by Galland and Scheibert (2013). The input parameters and uncertainties in both input and model results are presented in a clear and understandable way. The best-fit solution is a laccolith intrusion at 20 to 200 m depth. The magmatic overpressure assumed in the models is corroborated by calculations of the pressure increase due to conduit constriction during the eruption. The overall approach appears valid and is presented in a transparent way.

No action required.

Some of the figures are a bit small. I suggest to increase their size and the font size of labels. I also think that the Supplementary Fig. S5A should be added to the main manuscript text with some small adjustments (see below).

We appreciate this comment and have made all figures more readable, with additional adjustments as per the comments of Editor Plail.

*D. Appropriate use of statistics and treatment of uncertainties
The range of fitting models is described and uncertainties are discussed.*

No action required.

*E. Conclusions
The conclusions are valid and the transparent presentation of the uncertainties allows the reader to evaluate the robustness of the results and conclusions.
As the manuscript ends with the discussion (apart from the Methods section), I suggest to add a summarising sentence, a take-home message of some sort.*

*F. Suggested improvements
I have listed some minor comments in the table below.
Line Comment
43 Please replace precursor by prerequisite, as a precursor is a signal to most volcanologists. **Completed.***

45 The intrusion will often create a new vent, please mention.

We do not have evidence that the intrusion at Cordón Caulle created new vents. Indeed, the timing of the intrusion coincided with a period in the eruption when the active vent reduced in size. The surface manifestations of the intrusion included a broad area of fumarolic activity as well as numerous small-scale fault scarps. Although it is imaginable that in other cases such an intrusion may create a new vent, mentioning this does little to improve the clarity of the discussion about eruption-induced intrusions.

*52 I guess what the authors mean is not “rigid magma”, but the host rock or similar.
Please correct.*

We have changed to the wording to “highly viscous or solidified magma overhead”; “host rock” is not our intended message.

64 Please replace “m’s” by e.g. “metre-sized”

This has been corrected.

73 *The fractures mentioned in the text are hard to identify in Fig. 1. They are, however, nicely represented in the supplementary figure S5. I suggest adding Fig. S5A to Figure 1 and increase the thickness of the lines or the overall size of S5A.*

We appreciate this comment and have annotated Fig. 1 to show (in frame B) the fractures we speak of. We hesitate to increase the size and content of Fig. 1 by moving Fig. S5A into it, in part because the other reviewer also commented that some figures should be enlarged due to the details being hard to read. Instead, we cite Schipper et al (2013) in the text, as this is the first reference that identifies these ground cracks, and their Fig. 1a is a good high-resolution image of the ground deformation.

I find this satellite image a lot clearer to understand. The area of uplift in relation to the location of the vent is easier to see.

The cracks are now more clearly indicated in the Fig. 1B. As we cite on line 79 of the new manuscript, Schipper et al. (2013) recognized significant cracks around the vent. We refrain from moving the Fig. S5A into figure 1 as it will be too large and busy as a consequence of this change.

107 *Please add some references to articles on sill emplacement that support the suggestion that the initial sill was emplaced along a weak plane, e.g. Kavanagh et al., 2006, EPSL; Burchardt, 2008, JVGR.*

We have added two references to accommodate this suggestion: Burchardt, 2008, JVGR and Hogan et al., 1998 J. Struct. Geol. The Hogan et al reference is the first to suggest that sill emplacement occurs along weak planes.

117 *Change “m’s” to “m”. This has been corrected.*

124 *If the format of Nature Communication articles allows, I suggest to add a Discussion headline here to separate results from Discussion.*

This has been corrected; the headline “Discussion” has been added at the suggested place.

165 *Which part of the Supplementary Information does this statement refer to? It is unclear whether the authors indicate that the 2013 eruption was fed by the laccolith. More explanations are needed here or in the Supplementary Information.*

We appreciate this remark. We have adjusted this to read: “Supplementary Fig. 8”

I also suggest to add a reference to Sigmundsson et al. 2010 Nature on how a shallow sill was hit by a dyke resulting in the explosive part of the Eyjafjallajökull eruption in

2010.

We appreciate this suggestion, but do not see how this paper is relevant in the context of the discussion on late-staged explosions being fueled by a shallowly nested laccolith. It is important to remember that this paper is about eruptions causing intrusions, not intrusions resulting in eruptions. We therefore do not add this reference.

Figures Please consider increasing the size of most of the figures and the fonts within the figures. This has been corrected.

G. References

I suggest to add some references. See my comments under F. This has been corrected.

H. Clarity and context

The abstract is clear and appropriate. Introduction and conclusions are well written. I suggest to add a final statement (see above).

II. Reviewer 2 (Helge Gonnermann) comments:

The manuscript presents the discovery of shallow magma intrusion (laccolith) during the 2011 eruption of Cordon Caulle volcano in Chile. The work is based on remote sensing observations of surface deformation and of thermal emission, together with field observations. Overall this is an interesting discovery/hypothesis that is suitable for publication in Nature Communications. I therefore recommend the manuscript for publication. While I see no major flaws with the work presented, I do have some comments for consideration.

No action required.

1.) Figure 4: In my opinion it should be shown that a laccolith provides indeed the better match to the observed deformation compared to other deformation sources. For example, one could consider inflation of the upper conduit as a null hypothesis that needs to be ruled out. This should be easy to do, as there are analytical models for pipe-like inflation sources. This could be discussed within one or two sentences in the manuscript, with a brief discussion of the analysis deferred to the supplementary information.

We appreciate that other sources can lead to surface deformation, but in our search of the literature, we have yet to come across a paper or model involving a shallow cylindrical source that could plausibly explain a convex deformation field some 2.5 km in diameter and showing more than 200 m uplift. Deformation resulting from point source (Mogi), spheroid source, and

tensional faults at Cordón Caulle was the subject of work by Jay et al. (2014), whom we cite in our manuscript. These sources result in comparatively miniscule deformation (centimeters to decimeters). Similarly, models of inflation related to large overpressure (34-40 MPa) in subsurface dykes at Soufriere Hills, Montserrat predict surface uplift of only ~10cm, three orders of magnitude smaller than the measured uplift at Cordón Caulle. As the reviewer does not provide reference to this model nor further rationale as concerns the plausibility of such a source to produce a profound deformation field, we do not explore this further in the revised manuscript.

2.) Figure 1E: Could an alternate interpretation of this image be that snow has accumulated in local topographic lows (troughs) and that higher (more exposed) areas, such as small ridges/swells or steep slopes, have less/no snow cover? If that is not a viable alternative, perhaps add a brief sentence explaining why that is so?

We appreciate this question, but no, if one observes the topographic information conveyed on the maps, ie., both contour lines and color-coded elevations shown in the equally-scaled frames C and D, then there is no reason one should conclude that the snow-free zones are a consequence of high standing topography with steep slopes or preferential snow deposition in topographic lows. One can see plenty of snow-covered ground of equal elevation to the snow-free area and with the same physiographic character. Note also that the near-vent snow-free zone (mentioned in the caption of Fig. 1), correlates to the surface heat flow depicted in the Supplemental Figure 2, which we now reference in the revised figure 1 caption.

3.) Figure 2: These images are not easy to interpret. It would be helpful if the images were larger in size and if there was more annotation. What is the date on the left panel of Figure 2? It would be helpful to add a time line to this figure, indicating what the eruptive activity was. This would allow the reader to place the images into the overall context of the eruptive progression.

We appreciate these questions. We have enlarged all the figures according to this request, and a similar comment of the editor. We have added labels to signify the appearance of the lava flow. We have also added the statement that “the first three frames represent purely explosive activity, while the last three mark the beginning phases of hybrid explosive-effusive activity (Schipper et al., 2013)”, to the figure caption. To further link the radar imagery to the eruption chronology, we have improved the text in lines 83-92, now clearly indicating that the first three radar images capture the onset of deformation during explosive activity.

4.) I like Figure 3. Figure 3A is identical to Figure 1D and perhaps redundant, given likely space constraints. When I studied Figure 1, I was hoping for a difference map between 1D and 1C, which is shown in Figure 3C. I am wondering if it would make sense to combine Figures 1 and 3?

We appreciate this suggestion. For reasons we've already stated, namely that the figure 1 is already quite large and with a lot of conveyed information, we choose not to consolidate figures 1 and 3.

5.) Figure 5: I understand the objective for doing this modeling, but wonder whether it is actually necessary for the purpose of this manuscript. In fact, it potentially raises some detracting questions. For example, in Figure 4 it seems that the “best” model is for an intrusion depth of 90 m. According to Figure 5A that would seem to be above the level of fragmentation and imply intrusion of tephra. There are also a number of other questions about the conduit modeling that need to be given consideration, as there are a number of assumptions and uncertainties associated with such modeling (conduit size/shape, inlet pressure, exit conditions, magma rheology, volatile content, etc.). These, together with an assessment of uncertainty/sensitivity of model predictions, as well as a table of model parameters and parameter uncertainties, need to be presented in the supplementary information. Alternatively, one could perhaps defer a more thorough discussion of such modeling to a separate paper. In my opinion the paper does not require such modeling to make a convincing case for syn-eruptive intrusion.

While it is a fair point that the conduit model may not be “necessary”, we argue that it is important because it provides an independent, order-of-magnitude estimate of the overpressure in the conduit, which corroborates results from the deformation model, thus providing a mechanistic link between the eruption and the flexural deformation above the laccolith. We agree that an in-depth study of all possible conduit model variables is beyond the scope of the paper, but emphasize that two out of the three reviewers appreciate the modeling of conduit flow—one even finding it “more interesting” than the deformation model (Alessandro Bonforte)—and what it brings to the manuscript.

It is also very important to recognize that the deformation model solution shown in Fig. 4 is not the only permissible solution to the deformation. To clarify this, we have added the phrase “one, non-unique model solution” to the Fig. 4 caption. We also stated in the original manuscript that a rigorous exploration of the parameter space occurs by way of 200 separate model simulations in the supplementary information. This statement appears on lines 119-123 of the new manuscript. Note, some of these solutions imply laccolith depths >200 m, well below the fragmentation level. Therefore, the question of how the information in Fig. 4 could be combined and interpreted in the context of the CONFLOW model in Fig. 5, while valid, should also acknowledge that those two scenarios are not the only ones possible. Intrusion of tephra, is however, well documented in silicic systems—first identified by Heiken et al 1988 (cited in the manuscript). Therefore, we do not see this as a detracting question but one that perhaps prompts further productive thought.

6.) *Figure S7: It seems to me that the deformation modeling is the central part of this paper, because it is THE hypothesis test. Although not a necessary requirement, I am wondering whether it would not be sensible to provide a more in-depth presentation of the results of this analysis within the manuscript itself?*

We appreciate this suggestion. At the editor's request we have moved over many of the modeling details to the main manuscript methods section. We have also stated the selection of variables and limits of the Galland and Scheibert (2013) model in this Methods section. In addition, the sensitivity of the model to the different input parameters is provided in the supplementary Figure 7. As stated by reviewer 3: "Authors correctly show the dispersion of the parameters of 200 model processings."

Specifically, what I find lacking is an adequate presentation of model tradeoffs. For example, one could provide 3D graphs or 2D contour plots showing goodness of fit as a function of radius and pressure, as well as depth and pressure, and radius and depth for a given Young's modulus. This would indicate how well the model is resolved.

The "presentation of model tradeoffs" is exactly what is shown in the 3D graphs of Supplementary Figure 7. In these, goodness of fit is shown as a color code that maps on the 200 model solution points.

7.) *How is model "fit" defined?* **We appreciate this question. Fit is defined by the mismatch between the modeled and natural topography, ie., the goodness of fit. This is expressed as R^2 values, given in the figure caption (Fig. 4) and in the Supplementary Figure 7, where R^2 is shown by a color scale.**

8.) *Figure 4C: How is "natural topographic variations" defined? Is that the post-intrusion topographic profile minus the pre-intrusion profile?* **We have changed the wording to natural topography to clarify that it is a fit to the natural land surface, or topography.**

III. Reviewer 3 comments:

This is an interesting paper dealing about a particular and outstanding deformation measured during the eruption at Cordon caulle volcano, revealing a shallow subsurface dynamics of magma accompanying the eruptive event. Authors present a multidisciplinary investigation using data coming from different techniques. Data are well presented and discussed and the interpretation of the phenomenon fits the observation at surface. Authors perform also a model calculation even if, as they clearly and correctly state, the conditions are at the limits of the model assumptions; mainly due to the very shallow intrusion and the clearly not-elastic condition of the deformed medium. Personally, I think that a model is not always necessary for demonstrating an observed phenomenon, especially when the observations are so outstanding and the conditions are not so close to the theoretical assumptions;

sometimes, the uncertainties of the model are bigger than those associated to a simple direct interpretation by experts. In this case, however, "signal-to-noise" ratio is very high and this allows reliable results and Authors correctly show the dispersion of the parameters of 200 model processings. More interesting is (in my opinion) the conduit simulation in order to understand the pressure and depth of the intrusion process.

Manuscript is well written, I had only some difficulties in following lines from 78 to 85, trying to making order on the dates and the evolution of the deformation. Maybe a summary table or a simple timeline showing the most important dates of the observed phenomena and the images acquisitions could be useful to the reader. After this, i think that this paper deserves to be published. –Alessandro Bonforte

We appreciate the comment that the description of the radar imagery was not so clear. We have now updated the text on lines 83-92 to express more clearly the specific dates corresponding to key eruption phases, and that the first three radar images of Fig. 2 correspond to purely explosive activity. We note (line 87-88) that these images capture the onset of deformation. With the dates of each image clearly printed on the images themselves, we do not see the need for a separate figure or table indicating a timeline.